# Post-acute care for frail older people decreases 90-day emergency room visits, readmissions and mortality: An interventional study

Min-Chang Lee[1,2], Tai-Yin Wu[3,4,5]*, Sheng-Jean Huang[6,7], Ya-Mei Chen[8], Sheng-Huang Hsiao[9,10], Ching-Yao Tsai[5,11,12,13]

1 Center for Athletic Health Management, Renai Branch, Taipei City Hospital, Taipei, Taiwan, 2 Center for General Education, Taipei University of Marine Technology, Taipei, Taiwan, 3 Department of Family Medicine, Zhongxing Branch, Taipei City Hospital, Taipei, Taiwan, 4 Institute of Epidemiology and Preventive Medicine, National Taiwan University, Taipei, Taiwan, 5 General Education Center, University of Taipei, Taipei, Taiwan, 6 Taipei City Hospital, Taipei, Taiwan, 7 Department of Surgery, College of Medicine, National Taiwan University, Taipei, Taiwan, 8 Institute of Health Policy and Management, National Taiwan University, Taipei, Taiwan, 9 Department of Neurosurgery, Renai Branch, Taipei City Hospital, Taipei, Taiwan, 10 Department of Psychology, National Chengchi University, Taipei, Taiwan, 11 Department of Ophthalmology, Taipei City Hospital, Taipei, Taiwan, 12 Institute of Public Health, National Yang-Ming Chiao-Tung University, Taipei, Taiwan, 13 Department of Business Administration, Fu Jen Catholic University, New Taipei City, Taiwan

* taiyinwu@ntu.edu.tw

**Data Availability Statement:** All relevant data are within the paper and its Supporting Information files.

## Abstract

### Background

To evaluate the effects of post-acute care (PAC) on frail older adults after acute hospitalization in Taiwan.

### Methods

This was a multicenter interventional study. Frail patients aged $\geq$ 75 were recruited and divided into PAC or control group. The PAC group received comprehensive geriatric assessment (CGA) and multifactorial intervention including exercise, nutrition education, and medicinal adjustments for two to four weeks, while the control group received only CGA. Outcome measures included emergency room (ER) visits, readmissions, and mortality within 90 days after PAC.

### Results

Among 254 participants, 205 (87.6±6.0 years) were in the PAC and 49 (85.2±6.0 years) in the control group. PAC for more than two weeks significantly decreased 90-day ER visits (odds ratio [OR] 0.21, 95% confidence interval [CI] 0.10–0.43; p = 0.024), readmissions (OR 0.30, 95% CI 0.16–0.56; p < 0.001), and mortality (OR 0.20, 95% CI 0.04–0.87; p = 0.032). Having problems in self-care was an independent risk factor for 90-day ER visits (OR 2.11, 95% CI 1.17–3.78; p = 0.012), and having problems in usual activities was an independent

**Funding:** This study was supported by grant from Department of Health, Taipei City Government, Taipei, Taiwan (10901-62-067). The funders had no role in study design, data collection and analysis, decision to publish, or preparation of the manuscript.

**Competing interests:** The authors have declared that no competing interests exist.

**Abbreviations:** ADL, activities of daily living; CAM, Confusion Assessment Method; CFS, Clinical Frailty Scale; CGA, comprehensive geriatric assessment; CKD, chronic kidney disease; COPD, chronic obstructive pulmonary disease; EQ-5D, EuroQol-5 dimension; ER, emergency room; GDS-5, Geriatric Depression Scale-5 Item; HPAC, home-based post-acute care; IADL, instrumental activities of daily living; IPAC, inpatient-based post-acute care; MNA, mini Nutritional Assessment; PAC, post-acute care; SPMSQ, Short Portable Mental State Questionnaire; STEADI, Stop Elderly Accidents, Deaths, and Injuries.

risk factor for 90-day readmissions (OR 2.69, 95% CI 1.53–4.72; p = 0.001) and mortality (OR 3.16, 95% CI 1.16–8.63; p = 0.024).

## Conclusion

PAC program for more than two weeks could have beneficial effects on decreasing ER visits, readmissions, and mortality after an acute illness in frail older patients. Those who perceived severe problems in self-care and usual activities had a higher risk of subsequent adverse outcomes.

## Trial registration

ClinicalTrials.gov NCT Identifier: NCT05452395.

## Introduction

Frail older adults are vulnerable to a minor stress event [1]. They could experience tremendous health declines [2,3]. Previous studies have shown that frail older persons had higher risk of having hospitalization-associated disability than those without frailty [4–9].

Among frail individuals aged 70 or above, about 20% to 35% became disabled within one month after hospitalization [5]. Moreover, the risk of emergency room (ER) visits, readmissions, as well as mortality for the frail elderly was higher than those without frailty [10,11]. Both medical and social problems affect their clinical outcomes. Increasing evidence recommends that comprehensive geriatric assessment (CGA) and multifactorial intervention which includes exercise, nutrition, psychosocial program, and medical management could improve frailty status [12–16]. These multifactorial interventions are usually composed of frequent exercise training supplemented with other health managements such as nutritional program or medical review [12,13,15,17]. Importantly, these multifactorial interventions leading to lifestyle modification may reflect in decreased mortality and lessened medical resource utilization after the intervention.

However, most studies investigated community frail older adults in stable frail status [13,15,17]. This group could be different from frail older adults with hospitalization-associated disability after an acute illness. This population could be frailer and more disabled than their community counterparts.

Recent studies found that older adults with chronic conditions facing acute health crises benefited from both home-based and hospital-based integrated care on functional recovery [18–20]. After acute illness treatments, patients became medically stable and were in the post-acute status. They would stay hospitalized in a post-acute hospital ward. Taiwan has implemented post-acute care (PAC) for stroke since 2014 and for fragile fractures, traumatic neurological injury, and frail older people since 2017 [21]. There is hospital-based or home-based PAC for patients depending on patient's condition and families' capabilities. Recent studies have reported positive effects on functional recovery for patients with stroke [22–24] and fractures [25–27]. However, the PAC for frail older people is a unique design in Taiwan which includes CGA, multifactorial intervention, and home-based care [21]. Studies have shown the effectiveness of this PAC program for frail older people [28–31]. The prevalence of dementia, Parkinsonism, chronic obstructive pulmonary disease (COPD), or moderate chronic kidney disease (CKD) in elderly patients is high in Taiwan. Nevertheless, frail older adults with these diseases who were vulnerable to adverse outcomes and needed further care after

hospitalization were rarely investigated. Therefore, the aim of this study was to examine the effects of PAC program for frail older adults after hospitalization.

## Methods

### Study design

This was a multicenter interventional study with 90-day follow-up. We aimed to examine the effects of PAC program for frail older adults after hospitalization. We hypothesized that PAC program for frail older people may decrease 90-day emergency room visits, readmissions and mortality.

This study (S1 and S2 Files) was approved by the Research Ethics Committee (TCHIRB-10906008-E) and registered with Clinicaltrials.gov (NCT Identifier: NCT05452395). We followed the Transparent Reporting of Evaluations with Nonrandomized Designs (TREND) statement for reporting this paper. (S1 Checklist)

### Participants

Patients who participated in the PAC-frailty program in five hospitals in Taipei, Taiwan between September 1, 2017 and May 31, 2020 were enrolled. The follow-up was extended till August 31, 2020. Frailty was identified as mild to severe frailty using the Clinical Frailty Scale (CFS) [32]. The inclusion and exclusion criteria of PAC-frailty program were shown in Table 1. We divided the participants into two groups before being discharged from an acute hospital ward. Patients who participated in the PAC program, considered as intervention group, would be transitioned to their home for home-based (HPAC) or to a PAC hospital ward for inpatient-based (IPAC) model. The choice to receive either treatment was based on medical capacities, caregiver abilities, and patient willingness. Patients who were eligible but refused to receive PAC were considered the control group. To help minimize potential bias induced due to non-randomization, participants were matched with regards to age, gender and extent of disability. Most patients recruited were aware of the benefits of PAC to some extent and consented to join the program, hence resulting in few participants in the control group. Data of those who refused to participate in this study were not collected. Oral and written inform consents were obtained before study participation.

### Intervention group

**CGA.** At the beginning of PAC, CGA was performed at a PAC hospital. The CGA included the following. Participants were identified as mild (need assistances in high older

**Table 1. The screening criteria of post-acute care frailty program.**

| Inclusion criteria | Exclusion criteria |
| --- | --- |
| 1. Mild to severe frailty identified by the Clinical Frailty Scale<br>2. Age $\geq$ 75 years<br>3. Diagnosis with Parkinsonism, dementia, chronic obstructive pulmonary disease, or chronic kidney disease stage three or worse<br>4. Acute hospital stays between 3 to 30 days with deconditioning<br>5. Stable medical condition with no need of intensive care, laboratory examination, or oxygen dependence | 1. Refused to participate in the program<br>2. Candidate for other post-acute care programs (i.e. stroke, traumatic neurological injury, or fracture)<br>3. Unable to cooperate with the program due to mental or cognitive impairment<br>4. Long-term ventilator-dependence<br>5. Long-term bed-ridden status ($>$ 6 months)<br>6. Diagnosed as end of life and in need of palliative care<br>7. Diagnosed as major illness (i.e. end-stage renal disease) and in need of frequent inpatient treatment ($>$ 3 times over recent 6 months)<br>8. Institutional residents<br>9. Home medical care participants |

instrumental activities of daily living [IADL]), moderate (need help in activities of daily living [ADL]), or severe (complete dependence for personal care) frailty by CFS [32,33]. Functional status was measured with ADL (range 0–5; lower scores indicate more dependent) [34] and IADL (range 0–8; lower scores suggest more dependent) [35]. ADL dependence was defined as dependence of having a meal, toileting, bathing, dressing, and transferring from a chair. IADL dependence was defined as dependence of shopping, housework, food preparation, transportation, using telephone, laundry, and handling finances and medication. Fall risk was assessed using the Stop Elderly Accidents, Deaths, and Injuries (scoring as low, moderate, or high risk according to its algorithm) [36]. Cognitive function was evaluated by the Short Portable Mental State Questionnaire (range 0–10; scoring 0–2 imply normal, 3–4 mildly, 5–7 moderately, and 8–10 severely cognitive impairment) [37] and the Confusion Assessment Method (scoring as no confusion or confusion) [38]. Depression status was assessed by the Geriatric Depression Scale-5 Item (range 0–5; scoring $\geq$ 2 indicate depression) [39]. Nutritional status was assessed using the mini Nutritional Assessment Short Form (range 0–14; scoring 0–7 imply malnutrition, 8–11 at risk, and $\geq$ 12 normal) [40]. Quality of life was assessed with the EuroQol-5 dimension (scoring as no problem, some problem, or severe problem according to patient's answer) [41].

A transdisciplinary meeting was held to plan a personalized care program, focusing on managing geriatric syndromes and improving functional performances.

**PAC.** Patients in both PAC groups received similar services by well-trained geriatric rehabilitation teams. The inclusion criteria for both HPAC and IPAC groups were the same. These teams included geriatric physicians, nurses, physical therapists, occupational therapists, speech therapists, pharmacists, dietitians, social workers, and case managers. Physical therapy emphasized on strengthening exercise, endurance training, balance training, chest care, and ambulation training. Occupational therapy focused on independence in basic and instrumental ADL. Swallowing training, if needed, was provided by speech therapists for those patients with nasal-gastric tube insertion due to an acute swallowing disorder. Nutrition consultation was given by dietitians to patients and their caregivers helping them to prepare daily meals. Social workers provided social resources and welfare information when necessary. In addition, pharmacists played a pivotal role to adjust medication and prevent medication-associated functional decline. The case managers collaborated with team members and patients to fit the program in their daily lives and to enhance their health literacy and ability to self-management. The PAC duration was two to four weeks depending on individual condition.

Patients in the HPAC group received individualized rehabilitation programs at home by a physical therapist, occupational therapist, and/or speech therapist. The program included two to three sessions per week, 50 minutes per session, depending on individual condition. A home-based personalized rehabilitation program, which fitted their daily lives by using their home resources, was provided for patients and their caregivers. The program focused on strength training, ADL and IADL practice, basic mobility training, and reconditioning exercise. Also, individualized meal preparation was taught by a dietitian to the caregivers at home if necessary. The home environment was assessed and modified for those with a high risk of fall. Patients with cognitive impairment, depression, and delirium were referred to neurologists for appropriate treatment. Patients in the HPAC group were followed up at outpatient clinics to manage comorbidities and geriatric syndromes.

A hospital-based geriatric rehabilitation program was provided in the IPAC group at a PAC hospital ward. Rehabilitation was provided by physical therapist, occupational therapist, and/or speech therapist once to twice a day, 50 minutes per session on weekdays. If necessary, physician treated comorbidities and geriatric syndromes. Nursing care was provided for daily vital sign monitor. Other healthcare professionals were consulted during the PAC period as needed. There was no monetary incentive to increase compliance or adherence.

### Control group

Patients in the control group were provided with CGA at baseline and two weeks later by the same geriatric team. They were referred to outpatient's clinics for further treatment if needed. There was no other intervention.

### Outcome measurements

The primary outcomes were 90-day ER visits, readmissions, and mortality after PAC. The data were collected from their medical records. Demographics and clinical data including age, gender, medical diagnosis, duration of PAC, rehabilitation sessions, and living environment were also obtained from their medical records. Data were collected by a well-trained research assistant through standard protocols.

### Statistical analysis

The smallest unit that was being analyzed to assess intervention effects was individual participants. To compare the characteristics between groups, Chi-square test or independent t test was performed. Bivariate analyses were performed to examine the association between baseline characteristics and 90-days ER visits, readmissions, and mortality. Only significant variables were entered into multivariate analysis. Then, multivariate logistic regression analysis with full, forward, and backward methods was used to evaluate the odds ratio (OR) of the ER visits, readmissions, and mortality. The cumulative ER visits, readmissions, and mortality within 90 days after PAC were analyzed with Kaplan-Meier method. Log-rank test was used to evaluate the group differences. Between centers discordance was assessed using generalized estimation equation. SPSS version 20.0 (SPSS Inc., Chicago, USA) was used for all statistical analyses. A significant level was set at $p < 0.05$ using two-tailed test. Analysis was based on intention to treat.

## Results

### Baseline characteristics

A total of 1351 patients were screened for eligibility, 254 patients were recruited while 205 were in the PAC group and 49 were in the control group (Fig 1). Reasons for non-participation in the intervention group included self-reporting as non-frailty (n = 27), financial constraints (n = 6), living far away from the hospital (n = 10), or already being under other medical or rehabilitation care (n = 6). These patients were included in the control group.

Table 2 presented the baseline characteristics of the two groups. The mean age of participants was 87.6 ± 6.0 years for the PAC group and 85.2 ± 6.0 years for the control group (p = 0.13). The most prevalent diagnosis was dementia (40.5%) for the PAC group and CKD (73.5%) for the control group (p < 0.001). Only 14.2% participants in the PAC group were identified as mild frailty while 32.6% participants in the control group were identified as mild frailty (p = 0.009). Patients in the PAC group had more geriatric syndrome (cognitive impairment, depression, delirium, high risk for falls, and malnutrition) than those in the control group (p < 0.05). On average, the duration of PAC was 14.4 ± 5.4 days and 15.0 ± 10.6 sessions for the PAC group.

### ER visits, readmissions, and mortality

The mortality rate after PAC within 90 days was presented in Table 3. Factors associated with ER visits, readmissions, and mortality within 90 days were shown in S1–S3 Tables. The results of multivariate logistic regression were presented in Table 4A–4C. We observed that the

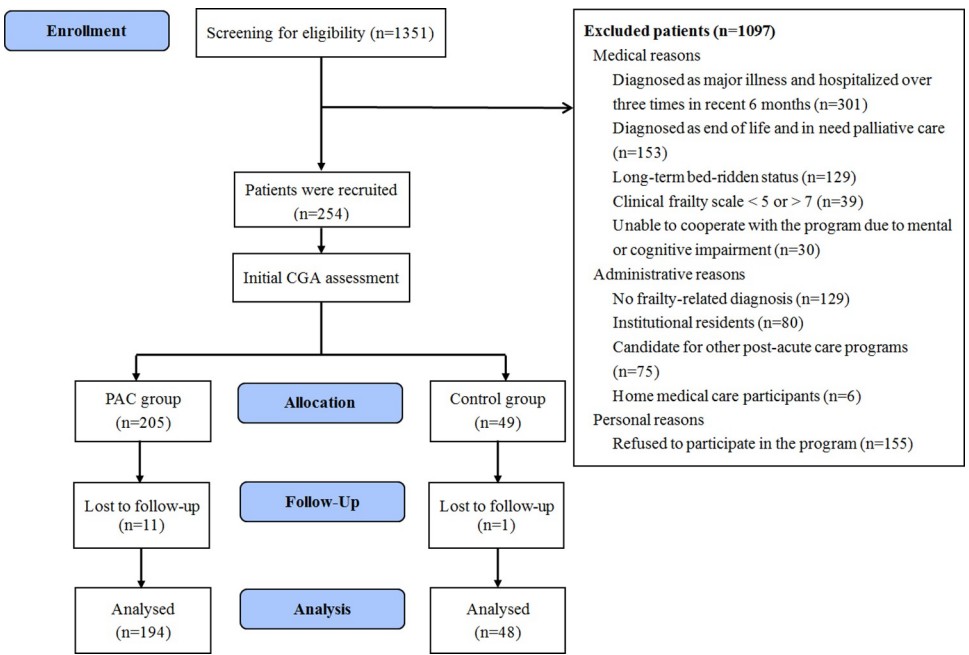

**Fig 1. CONSORT flow diagram.** Note: CGA, comprehensive geriatric assessment; PAC, post-acute care.

duration of PAC was an independent protecting factor for ER visits (OR 0.21, 95% confidence interval [CI] 0.10–0.43; p = 0.024), readmissions (OR 0.30, 95% CI 0.16–0.56; p < 0.001), and mortality (OR 0.20, 95% CI 0.04–0.87; p = 0.032). Baseline severe problem in self-care was an independent risk factor for ER visits (OR 2.11, 95% CI 1.17–3.78; p = 0.012). Severe problem in usual activities was an independent risk factor for readmissions (OR 2.69, 95% CI 1.53–4.72; p = 0.001) and mortality (OR 3.16, 95% CI 1.16–8.63; p = 0.024). Patients with Parkinsonism had lower risk of readmissions (OR 0.15, 95% CI 0.04–0.54; p = 0.004).

Compared with the controls (Fig 2A–2C), the PAC group had significant lower cumulative incidences in ER visits (p = 0.031) and mortality (p = 0.014) within 90 days after PAC. There was no significant difference in practice between centers with regards to readmissions and mortality.

## Discussion

We observed that frail patients receiving PAC had significantly decreased ER visits and mortality within 90 days compared with the control group. In addition, PAC for more than two weeks had positive effects on decreasing 90-day ER visits, readmissions, and mortality. At baseline, severe problems in self-care was a risk factor for ER visits, and severe problems in usual activities was a risk factor for readmissions and mortality.

Several studies indicated that older patients had difficulties in daily activities and outpatient visiting after hospitalization [9], and frail older patients had higher subsequent medical resource utilization and mortality than those without frailty [10,11]. Therefore, continuous care after hospitalization was suggested to prevent adverse outcomes [11]. In our study, the PAC program could significantly lower ER visits and mortality within 90 days after PAC. The underlying cause why PAC program helps decrease ER visits remains speculative. Improvements in delirium was noted in the intervention group which might lower ER visits. Also, transdisciplinary intervention improves health literacy and enables family and patient alike to

**Table 2. Baseline characteristics of the participants (n = 254).**

| Characteristics | PAC group (n = 205) | Control (n = 49) | *p* value |
|---|---|---|---|
| **Age (years)** | 87.6±6.0 | 85.2±6.0 | 0.13 |
| **Gender (men)** | 101 (49.3%) | 28 (57.1%) | 0.32 |
| **Medical diagnosis** | | | <0.001 |
| Dementia | 83 (40.5%) | 1 (2.0%) | |
| Parkinsonism | 20 (9.8%) | 5 (10.2%) | |
| Chronic kidney disease | 70 (34.1%) | 36 (73.5%) | |
| Chronic obstructive pulmonary disease | 32 (15.6%) | 7 (14.3%) | |
| **Frailty (CFS)** | | | 0.009 |
| Mild | 29 (14.2%) | 16 (32.6%) | |
| Moderate | 72 (35.1%) | 14 (28.6%) | |
| Severe | 104 (50.7%) | 19 (38.8%) | |
| **Duration of PAC / observation (days)** | 14.4±5.4 | 15.3±3.6 | 0.29 |
| **Rehabilitation sessions** | 15.0±10.6 | - | |
| **ADL dependence** | 28 (13.7%) | 5 (10.2%) | 0.52 |
| **IADL dependence** | 7 (3.4%) | 0 (0.0%) | 0.35 |
| **Cognition (SPMSQ)** | | | 0.006 |
| Normal | 52 (26.8%) | 22 (44.9%) | |
| Mild impairment | 82 (42.3%) | 8 (16.3%) | |
| Moderate impairment | 33 (17.0%) | 12 (24.5%) | |
| Severe impairment | 27 (13.9%) | 7 (14.3%) | |
| **Depression (GDS)** | 67 (32.7%) | 5 (10.2%) | 0.002 |
| **Delirium (CAM)** | 83 (42.3%) | 1 (2.1%) | <0.001 |
| **Fall risk (STEADI)** | | | <0.001 |
| Low risk | 12 (6.0%) | 14 (28.6%) | |
| Moderate risk | 48 (23.9%) | 11 (22.4%) | |
| High risk | 141 (70.1%) | 24 (49.0%) | |
| **Nutritional status (MNA)** | | | <0.001 |
| Normal | 18 (8.9%) | 14 (29.8%) | |
| At risk of malnutrition | 105 (52.0%) | 26 (55.3%) | |
| Malnutrition | 79 (39.1%) | 7 (14.9%) | |
| **EuroQol-5D** | | | |
| Mobility | | | 0.13 |
| No problem | 17 (8.8%) | 8 (16.3%) | |
| Some problem | 106 (54.6%) | 20 (40.8%) | |
| Severe problem | 71 (36.6%) | 21 (42.9%) | |
| Self-care | | | 0.30 |
| No problem | 25 (12.9%) | 6 (12.3%) | |
| Some problem | 93 (47.9%) | 18 (36.7%) | |
| Severe problem | 76 (39.2%) | 25 (51.0%) | |
| Usual activities | | | 0.08 |
| No problem | 16 (8.2%) | 7 (14.3%) | |
| Some problem | 97 (50.0%) | 16 (32.7%) | |
| Severe problem | 81 (41.8%) | 26 (53.0%) | |
| Pain / discomfort | | | 0.14 |
| No problem | 86 (44.3%) | 15 (30.6%) | |
| Some problem | 88 (45.4%) | 30 (61.2%) | |

(*Continued*)

**Table 2.** (Continued)

| Characteristics | PAC group (n = 205) | Control (n = 49) | p value |
|---|---|---|---|
| Severe problem | 20 (10.3%) | 4 (8.2%) | |
| Anxiety / depression | | | 0.86 |
| No problem | 104 (53.9%) | 28 (57.1%) | |
| Some problem | 71 (36.8%) | 18 (36.7%) | |
| Severe problem | 18 (9.3%) | 3 (6.1%) | |
| Living environment (Ground floor) | 40 (19.5%) | 16 (32.7%) | 0.046 |

Abbreviations: ADL, activities of daily living; CAM, Confusion Assessment Method; CFS, Clinical Frailty Scale; EQ-5D, EuroQol-5 dimension; GDS, Geriatric Depression Scale; IADL, instrumental activities of daily living; MNA, Mini Nutrition Assessment; PAC, post-acute care; SPMSQ, Short Portable Mental Status Questionnaire; STEADI, Stop Elderly Accidents, Deaths, and Injuries.

manage minor symptoms so exempting ER visits. A randomized clinical trial investigated 4 weeks of intervention combining nutrition supplement and exercise rehabilitation for older adults discharged from acute illness reported no significant difference in readmission rates between intervention and control groups [42]. Another study reported the duration of exercise intervention for community-dwelling frail older people lasting at least 2.5 months was helpful [43]. This may indicate that frail older adults may need longer continuous care to benefit from intervention for lower readmission risk.

We observed that the participants who received PAC for more than two weeks had lower risk of ER visits, readmissions and mortality. Although the duration was short, these participants received CGA with multifactorial interventions and intensive rehabilitation in both IPAC and HPAC groups. In our study, we excluded those who were hospitalized over three times in recent 6 months which meant our participants were confronting a new acute health crisis. A complete program combining CGA and multifactorial interventions might help frail older adults live a healthy lifestyle which may decrease medical resource utilization. Recent evidence suggested patient education, self-management support, and rehabilitation were key elements to keep patients and families running personalized care programs after PAC [44]. Our PAC program did not only provide rehabilitation services but also promote patients and families' knowledge of self-care and empower them to cope with their care needs/illness/frailty better. This might probably be the explanation for the findings we observed.

In the present study, frail older adults with severe problems in self-care and usual activities according to EQ-5D categories at baseline had a higher risk of adverse outcomes than those without them. These patients may have a high possibility to be totally dependent on ADL after discharging home which resulted in subsequent adverse outcomes. Studies have found that older adults with hospitalization-associated disability at acute discharge would have higher risk in readmission and mortality than those without it [45,46]. Unmet needs at discharge have been a strong predictor for readmission [47–49]. Therefore, it is important to identify these frail older patients who perceived themselves as having severe problems in self-care and usual activities before discharge from acute hospitalization. A good transition from acute care

**Table 3. Mortality rate after PAC within 90 days (n = 254).**

| Mortality | PAC group (n = 205) | Control (n = 49) | p value |
|---|---|---|---|
| 30 days | 7 (3.4%) | 5 (10.2%) | 0.044 |
| 60 days | 10 (4.9%) | 6 (12.2%) | 0.057 |
| 90 days | 12 (5.9%) | 8 (16.3%) | 0.014 |

**Table 4. A. Multivariate model of the factors associated with emergency room visits within 90 days (n = 254).** B. Multivariate model of the factors associated with readmission within 90 days (n = 254). C. Multivariate model of the factors associated with mortality within 90 days (n = 254).

| Variables | Model 1: full model | | Model 2:forward selection | | Model 3:backward selection | |
|---|---|---|---|---|---|---|
| | OR (odds ratio) | p | OR (odds ratio) | p | OR (odds ratio) | p |
| PAC | 1.12 (0.52–2.42) | 0.77 | - | - | - | - |
| Duration of PAC (≧15 days) | 0.25 (1.11–8.35) | <0.001 | 0.21 (0.10–0.43) | 0.024 | 0.21 (0.10–0.43) | 0.024 |
| Dementia | 0.64 (0.31–1.30) | 0.22 | - | - | - | - |
| Delirium (CAM) | 0.69 (0.34–1.41) | 0.31 | - | - | - | - |
| Severe problem in self-care (EQ-5D) | 1.69 (0.71–3.99) | 0.23 | 2.11 (1.17–3.78) | 0.012 | 2.11 (1.17–3.78) | 0.012 |
| Severe problem in usual activities (EQ-5D) | 1.48 (0.64–3.46) | 0.36 | - | - | - | - |
| Parkinsonism | 0.16 (0.04–0.58) | 0.006 | 0.15 (0.04–0.54) | 0.004 | 0.15 (0.04–0.54) | 0.004 |
| Duration of PAC (≧15 days) | 0.29 (0.15–0.54) | <0.001 | 0.30 (0.16–0.56) | <0.001 | 0.30 (0.16–0.56) | <0.001 |
| Severe frailty (CFS) | 1.61 (0.85–3.03) | 0.14 | - | - | - | - |
| Severe problem in self-care (EQ-5D) | 1.08 (0.47–2.51) | 0.86 | - | - | - | - |
| Severe problem in usual activities (EQ-5D) | 2.09 (0.92–4.74) | 0.08 | 2.69 (1.53–4.72) | 0.001 | 2.69 (1.53–4.72) | 0.001 |
| PAC | 0.54 (0.20–1.50) | 0.24 | - | - | - | - |
| Duration of PAC (≧15 days) | 0.25 (0.05–1.16) | 0.08 | 0.20 (0.04–0.87) | 0.032 | 0.20 (0.04–0.87) | 0.032 |
| Severe problem in usual activities (EQ-5D) | 3.05 (1.11–8.35) | 0.030 | 3.16 (1.16–8.63) | 0.024 | 3.16 (1.16–8.63) | 0.024 |

Abbreviations: CAM, Confusion Assessment Method; EQ-5D, EuroQol-5 dimension; PAC, Post-acute care; CFS, Clinical Frailty Scale; EQ-5D, EuroQol-5 dimension; PAC, Post-acute care; EQ-5D, EuroQol-5 dimension; PAC, Post-acute care.

to post-acute care is necessary for them to prevent being totally dependent on ADL. However, some of these patients may experience very severe frailty after acute illness. They might need to be transferred to palliative care rather than intensive rehabilitation such as PAC. To provide appropriate transition care, careful evaluation and discharge planning is needed before discharge from acute hospitalization.

We also found that patients with Parkinsonism had a lower risk of readmissions as compared to patients with COPD, CKD, and dementia. They might have lower risks of acute health crises. Also, they were more aware of the importance of rehabilitation so that their exercise compliance might be better than other patients.

There are several study characteristics. This was a multicenter study. We enrolled frail older patients in five hospitals in Taipei. In addition, we recruited these frail older patients with specific underlying disabling diseases that were rarely discussed in previous studies. Besides, we tracked ER visits, readmissions, and mortality within 90 days after PAC. The loss to follow-up

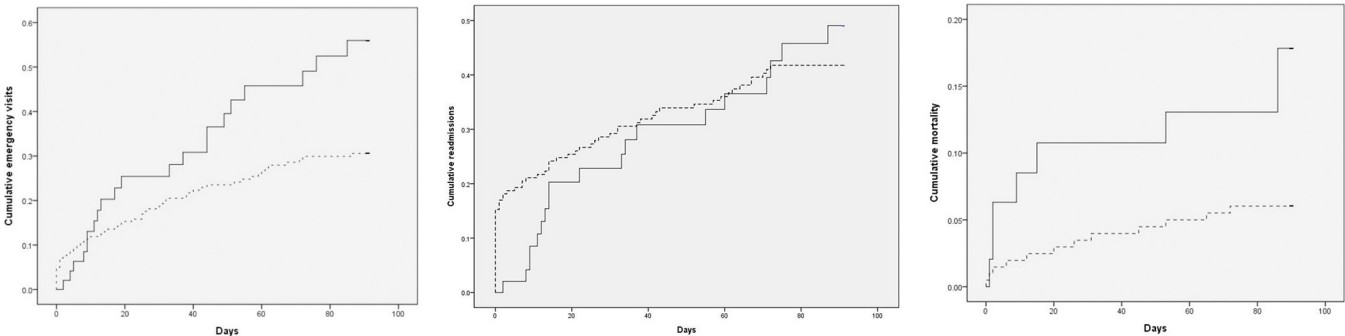

**Fig 2.** A. Cumulative emergency room visits within 90 days after PAC intervention. B. Cumulative readmissions within 90 days after PAC intervention. C. Cumulative mortality within 90 days after PAC intervention.

rate was minimal (5.3% in the PAC and 2.0% in the control group). We included subjective outcomes such as EQ-5D in our CGA which could significantly affect the subsequent adverse outcomes after PAC. This was rarely discussed in previous studies. The outcomes were significant after only 90 days of follow up.

This study has some limitations. There were few participants in our study. This was not a randomized controlled trial. There were some differences in baseline characteristics between the two groups. The willingness to participate in the services would be a confounder. However, randomization in clinical settings could have to some extent ethical concerns. Also, even though the control group had better function at baseline than the PAC group, good quality of care may be the key element to reduce adverse outcomes [28]. We recorded all-cause ER visits, readmissions, and mortality and there was no further analysis for the underlying reasons. Also, we were not able to know whether participants visit or admit at other hospitals or not.

## Conclusions and implications

In conclusion, PAC over two weeks could decrease 90-day ER visits, readmissions, and mortality for frail older patients after an acute illness. Besides, it is important to identify those who reported severe problems in self-care and usual activities to prevent further adverse outcomes. We recommend that patients with chronic diseases, particularly Parkinsonism, might benefit most from the PAC program. Large-scale and long-term functional gains and quality of life after PAC is our next focus of research.

## Supporting information

**S1 Checklist. Transparent Reporting of Evaluations with Nonrandomized Designs (TREND) statement.**
(DOCX)

**S1 Table. Univariate analysis of the factors associated with emergency room visits within 90 days (n = 254).**
(DOCX)

**S2 Table. Univariate analysis of the factors associated with readmission within 90 days (n = 254).**
(DOCX)

**S3 Table. Univariate analysis of the factors associated with mortality within 90 days (n = 254).**
(DOCX)

**S1 File. Original clinical research protocol.**
(DOCX)

**S2 File. Translated clinical research protocol.**
(DOCX)

**S3 File. Minimal data set.**
(XLSX)

## Acknowledgments

We acknowledge the assistants Guo-Shou Wang, Tzu-Heng Fu, Po-Yu Huang, and I-Xuan Chen for their kind help in data collection, data analysis and managerial support.

## Author Contributions

**Conceptualization:** Tai-Yin Wu, Sheng-Jean Huang.

**Data curation:** Min-Chang Lee.

**Formal analysis:** Min-Chang Lee, Tai-Yin Wu.

**Funding acquisition:** Tai-Yin Wu.

**Investigation:** Min-Chang Lee.

**Methodology:** Sheng-Jean Huang.

**Project administration:** Min-Chang Lee.

**Resources:** Sheng-Huang Hsiao, Ching-Yao Tsai.

**Supervision:** Tai-Yin Wu.

**Writing – original draft:** Min-Chang Lee.

**Writing – review & editing:** Tai-Yin Wu, Ya-Mei Chen.

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
