## [Decision Letter · Decision Letter 0]

18 Aug 2022

PONE-D-22-18127Post-acute care for frail older people decreases 90-day emergency room visits, readmissions and mortality: an interventional studyPLOS ONE

Dear Dr. Wu,

Thank you for submitting your manuscript to PLOS ONE. After careful consideration, we feel that it has merit but does not fully meet PLOS ONE’s publication criteria as it currently stands. Therefore, we invite you to submit a revised version of the manuscript that addresses the points raised during the review process.

We look forward to receiving your revised manuscript.

Kind regards,

Yoshihiro Fukumoto

Academic Editor

PLOS ONE

Journal Requirements:

 "This study was supported by grant from Department of Health, Taipei City Government, Taipei, Taiwan (10901-62-067)."

Reviewers' comments:

Reviewer's Responses to Questions

**Comments to the Author**

1. Is the manuscript technically sound, and do the data support the conclusions?

Reviewer #1: No

Reviewer #2: Yes

2. Has the statistical analysis been performed appropriately and rigorously? 

Reviewer #1: I Don't Know

Reviewer #2: Yes

3. Have the authors made all data underlying the findings in their manuscript fully available?

Reviewer #1: No

Reviewer #2: No

4. Is the manuscript presented in an intelligible fashion and written in standard English?

Reviewer #1: Yes

Reviewer #2: Yes

5. Review Comments to the Author

Reviewer #1: The authors have reported the dramatic effects of a post-acute care (PAC) program, which was carried out a small period of 2 to 4 weeks in hospital or at home, on decreasing 90-day emergency room visits, 90-day readmissions, and mortality within 90 days after the PAC in a multicenter interventional study of Taiwanese older 254 patients with frailty. They have sought to demonstrate the dramatic effects via very low odds ratios of the three primary outcomes in the interventional study; however, I have doubts about the propriety of the statistical analysis because this study was not a retrospective study. Mortality (i.e., mortality rate) is generally expressed as a numerical value in units of deaths per individuals; how did the authors calculate the very low odds ratio of the mortality of the study participants?

Even if the statistical analysis was appropriate, the dramatic effects should be demonstrated with data of more study participants. In addition, the authors should explain the PAC program in more detail.

Reviewer #2: Overall, your study is straightforward, and your results are clearly reported. However, your methodology is vague and more details are needed in order to understand both the intervention and the assessment strategy. Please consider the following:

Abstract

Methods – Readers are left wondering what a multifactorial intervention is. You haven’t described it in detail either here or in the body of your paper.

Conclusion – Here you say that a program for 2 weeks could be beneficial but your results are that the program needs to be more than 2 weeks.

Introduction

Your argument in support of your study is adequate but could be improved with formal editing to smooth out the wording. Switching between present and past tense is somewhat confusing and makes your logic difficult to follow.

Line 74 – It would be helpful to provide a bit more detail regarding the multifactorial interventions your citing here. That would allow readers to compare them with your intervention.

Line 76 – What do you mean by “other health managements?” Please be specific.

Line 86 – Please define “post-acute” for readers. Also, how is hospital-based PAC provided? Do patients return to the hospital for the program on a daily basis or stay admitted to the hospital? “Post-acute” infers that they are discharged from an acute hospital, but in that case, where is the hospital-based program delivered? A lot of detail isn’t needed here in the Introduction but is badly needed in your Methods section.

Lines 90-96 – What is the unique design of the PAC for frail older people? Since this is apparently the rationale for your present study, you need to be very clear on that point. Your reasoning here is a bit difficult to follow. You’ve cited multiple studies reporting the outcomes of this program in the same population but then you’ve somewhat arbitrarily identified a series of chronic diseases that you say are understudied. Is there some reason that this population is particularly notable and in need of study? It’s not clear why you cited the Medicare Payment Advisory Commission in support of this statement.

Methods

You have not adequately described either the intervention or your assessment (measurements).

Study design – Your inclusion/exclusion criteria should be reported in the body of your paper, not as a supplementary document, where some readers may not have access to them.

Participants – So some participants remained as inpatients during the intervention? How were they differentiated from “acute” for delivery of “post-acute” care?

Intervention (CGA) – You seem to be describing the PAC instead of the CGA. I suggest describing the assessment separately, and then describing the intervention (PAC) separately. Also, what do you mean by “smooth transition to home” after PAC (line 132)? Isn’t the program delivered after discharge? Please explain where and when the assessment (CGA) and intervention (PAC) were delivered.

Intervention (PAC) – How long was the program? Your Abstract reported 2-4 weeks, but the average time reported in your Results was only 14 days (2 weeks). Did any participants complete 4 weeks? And if the control was assessed at 2 weeks, would it be a fair comparison to a 4 week intervention? Please describe the intervention in detail. Based on your identification of the therapists, it sounds like a standard rehab program. In particular, what did the speech therapists do? These are frail older adults, not stroke patients. What was the role of speech therapy? Your description of the hospital-based rehabilitation program also sounds like a standard rehab program, but if patients were hospitalized with daily physician rounds, nursing care, etc., how was this “post-acute?”

Outcome measurements – Please describe the scoring conventions for all of your outcomes. For measures that were categorized for your results (for example frailty was categorized as mild, moderate, severe), please explain how the different categories were determined or calculated. Also, if there are validated Chinese versions of these tools that you used, please cite them.

Results

The participants in your program were quite old, well within the old-old category of aging. I suggest adding their mean age to your Abstract to emphasize the effectiveness of your program.

The data in Table S2 are not effectively reported. What do the variables on the Y-axis represent? For example, how is “frailty” defined? What was improved? Are you saying that 31% of the intervention and 37% of the control improved their frailty in some way? That doesn’t seem to demonstrate that your program was very effective.

Table 1 – Please explain either here or in your Methods what the variables along the Y-axis represent. For example, how were mild, moderate, and severe frailty defined? And it appears that on average, most participants continued for the minimum length of the program. If the program was offered for 2-4 weeks, why didn’t they continue for the full 4 weeks? And if the standard deviation for the intervention was 5 days, it appears that some participants didn’t continue for even 2 weeks. Please clarify.

Discussion

It’s difficult to follow your Discussion without a clear and detailed description of the intervention. For example, you say that “transdisciplinary intervention” (line 270) improves health literacy and ability to self-manage, but it’s not clear if your program addressed either of those variables.

Line 271 – What do you mean by “a similar observation?” Similar to what? Did your intervention include nutrition and exercise? You haven’t described your intervention in enough detail to allow readers to compare it to the studies you’re discussing here.

Line 282 – Here you refer back to your exclusion criteria but readers may not have access to your supplemental files. This is an example of why they need to be reported in the body of your paper instead of as supplementary material.

Lines 288-290 – Here you make a point about what your program provided but you didn’t describe it as part of your Methods.

Lines 292-293 – Please define “severe problems” as part of your Methods or your Results.

Line 298 – I think you mean “identify” instead of “find out.” You’re making a very important point here, so you want your meaning to be clear. Also, this point may be something to consider as part of your Conclusion.

Line 305 – Your point in this sentence isn’t clear. How can you provide evaluation and discharge planning before acute hospitalization? Do you mean before discharge from acute hospitalization? Please clarify.

Paragraph beginning on line 310 – The study characteristics you identify here should be clearly reported in your Methods. Some of them are either not clear or missing altogether.

Line 321 – Excellent point about ethical concerns and randomization.

Tables – Please go through all of your tables to ensure that each categorical variable along the Y-axis is clearly defined.

6. PLOS authors have the option to publish the peer review history of their article (what does this mean?). If published, this will include your full peer review and any attached files.

Reviewer #1: No

Reviewer #2: No

---

## [Author Response · Author response to Decision Letter 0]

4 Nov 2022

PONE-D-22-18127

Post-acute care for frail older people decreases 90-day emergency room visits, readmissions and mortality: an interventional study

Dear Dr. Yoshihiro Fukumoto:

Sep 30, 2022

The authors appreciate deeply your kind considerations and for giving us the opportunity to revise our manuscript. We have revised the manuscript in a point-by-point manner addressing your comments. All the changes in the revised unmarked manuscript are in red color.

Sincerely,

Min-Chang Lee, Tai-Yin Wu, Sheng-Jean Huang, Ya-Mei Chen, Sheng-Huang Hsiao, Ching-Yao Tsai 

 

REVIEWER 1:

Thank you for your insightful suggestions. Please find our responses to your comments below in point-by-point fashion.

[Comment 1]

The authors have reported the dramatic effects of a post-acute care (PAC) program, which was carried out a small period of 2 to 4 weeks in hospital or at home, on decreasing 90-day emergency room visits, 90-day readmissions, and mortality within 90 days after the PAC in a multicenter interventional study of Taiwanese older 254 patients with frailty. They have sought to demonstrate the dramatic effects via very low odds ratios of the three primary outcomes in the interventional study; however, I have doubts about the propriety of the statistical analysis because this study was not a retrospective study. Mortality (i.e., mortality rate) is generally expressed as a numerical value in units of deaths per individuals; how did the authors calculate the very low odds ratio of the mortality of the study participants?

[Reply]

Thank you for the comment. We used multivariate logistic regression analysis with full, forward, and backward methods to evaluate the odds ratio (OR) of the mortality. We have added the mortality rate according to your comment and please refer to Line 238; and Page 18, Table 3:

“The mortality rate after PAC within 90 days was presented in Table 3.”

[Comment 2]

Even if the statistical analysis was appropriate, the dramatic effects should be demonstrated with data of more study participants.

[Reply]

We appreciate the reviewer’s comments and have added a statement addressing this concern in the study limitation section. Please refer to line 328, and 341-342:

“There were few participants in our study.”

“Large-scale and long-term functional gains and quality of life after PAC is our next focus of research.”

[Comment 3]

In addition, the authors should explain the PAC program in more detail.

[Reply]

We have appreciated the reviewer’s comments and addressed the PAC program in detail. Please refer to line 151-165:

“PAC

Patients in both PAC groups received similar services by well-trained geriatric rehabilitation teams. The inclusion criteria for both HPAC and IPAC groups were the same. These teams included geriatric physicians, nurses, physical therapists, occupational therapists, speech therapists, pharmacists, dietitians, social workers, and case managers. Physical therapy emphasized on strengthening exercise, endurance training, balance training, chest care, and ambulation training. Occupational therapy focused on independence in basic and instrumental ADL. Swallowing training, if needed, was provided by speech therapists for those patients with nasal-gastric tube insertion due to an acute swallowing disorder. Nutrition consultation was given by dietitians to patients and their caregivers helping them to prepare daily meals. Social workers provided social resources and welfare information when necessary. In addition, pharmacists played a pivotal role to adjust medication and prevent medication-associated functional decline. The case managers collaborated with team members and patients to fit the program in their daily lives and to enhance their health literacy and ability to self-management. The PAC duration was two to four weeks depending on individual condition.”

 

REVIEWER 2:

We appreciate your positive view on this study. We have responded to your comments below.

Abstract

[Comment 1]

Methods – Readers are left wondering what a multifactorial intervention is. You haven’t described it in detail either here or in the body of your paper.

[Reply]

We appreciate the reviewer’s comments and have added another statement addressing this concern in line 28-30:

“The PAC group received comprehensive geriatric assessment (CGA) and multifactorial intervention including exercise, nutrition education, and medicinal adjustments for two to four weeks, while the control group received only CGA.”

[Comment 2]

Conclusion – Here you say that a program for 2 weeks could be beneficial but your results are that the program needs to be more than 2 weeks.

[Reply]

We have revised according to your comment. Please refer to line 41-42:

“PAC program for more than two weeks could have beneficial effects on decreasing ER visits, readmissions, and mortality after an acute illness in frail older patients.”

Introduction

[Comment 1]

Your argument in support of your study is adequate but could be improved with formal editing to smooth out the wording. Switching between present and past tense is somewhat confusing and makes your logic difficult to follow.

[Reply]

We appreciate the reviewer’s comments and have revised our words. Please refer to line 74-77, 91-95, and 98-100:

“Both medical and social problems affect their clinical outcomes. Increasing evidence recommends that comprehensive geriatric assessment (CGA) and multifactorial intervention which includes exercise, nutrition, psychosocial program, and medical management could improve frailty status [12-16].”

“There is hospital-based or home-based PAC for patients depending on patient’s condition and families’ capabilities. Recent studies have reported positive effects on functional recovery for patients with stroke [22-24] and fractures [25-27]. However, the PAC for frail older people is a unique design in Taiwan which includes CGA, multifactorial intervention, and home-based care [21].”

“Nevertheless, frail older adults with these diseases who were vulnerable to adverse outcomes and needed further care after hospitalization were rarely investigated.”

[Comment 2]

Line 74 – It would be helpful to provide a bit more detail regarding the multifactorial interventions your citing here. That would allow readers to compare them with your intervention.

[Reply]

Thank you for your comment. We have added the detail regarding the multifactorial interventions. Please refer to line 74-77:

“Increasing evidence recommends that comprehensive geriatric assessment (CGA) and multifactorial intervention which includes exercise, nutrition, psychosocial program, and medical management could improve frailty status [12-16].”

[Comment 3]

Line 76 – What do you mean by “other health managements?” Please be specific.

[Reply]

We appreciate your comment. We have added the detail regarding the multifactorial intervention in Line 74-77. This multifactorial intervention included exercise training, nutritional adjustment, psychosocial program, and medical reviews. Among different studies, we found that there was usually exercise training in multifactorial intervention along with other health managements.

We have revised the sentence and please refer to Line 77-79:

“These multifactorial interventions are usually composed of frequent exercise training supplemented with other health managements such as nutritional program or medical review [12, 13, 15, 17].”

[Comment 4]

Line 86 – Please define “post-acute” for readers. Also, how is hospital-based PAC provided? Do patients return to the hospital for the program on a daily basis or stay admitted to the hospital? “Post-acute” infers that they are discharged from an acute hospital, but in that case, where is the hospital-based program delivered? A lot of detail isn’t needed here in the Introduction but is badly needed in your Methods section.

[Reply]

Thanks for the comments. After acute illness treatments, patients become medically stable and are in post-acute status. Patients would discharge from an acute hospital ward and transfer to a post-acute hospital ward near their home for further rehabilitation and care for those who choose hospital-based PAC in Taiwan. We have addressed reviewer’s concerns in line 88-89:

“After acute illness treatments, patients became medically stable and were in the post-acute status. They would stay hospitalized in a post-acute hospital ward.”

[Comment 5]

Lines 90-96 – What is the unique design of the PAC for frail older people? Since this is apparently the rationale for your present study, you need to be very clear on that point. Your reasoning here is a bit difficult to follow. You’ve cited multiple studies reporting the outcomes of this program in the same population but then you’ve somewhat arbitrarily identified a series of chronic diseases that you say are understudied. Is there some reason that this population is particularly notable and in need of study? It’s not clear why you cited the Medicare Payment Advisory Commission in support of this statement.

[Reply]

We appreciate reviewer’s comments and have addressed reviewer’s concerns in line 94-100:

“However, the PAC for frail older people is a unique design in Taiwan which includes CGA, multifactorial intervention, and home-based care [21]. Studies have shown the effectiveness of this PAC program for frail older people [28-31]. The prevalence of dementia, Parkinsonism, chronic obstructive pulmonary disease (COPD), or moderate chronic kidney disease (CKD) in elderly patients is high in Taiwan. Nevertheless, frail older adults with these diseases who were vulnerable to adverse outcomes and needed further care after hospitalization were rarely investigated.”

Methods

[Comment 1]

You have not adequately described either the intervention or your assessment (measurements).

[Reply]

Thanks for your comments. We have addressed this concern in the Intervention (CGA), Intervention (PAC), and outcome measurements. Please refer to line 130-150, 151-165, and 189-194:

“CGA

At the beginning of PAC, CGA was performed at a PAC hospital. The CGA included the following. Participants were identified as mild (need assistances in high older instrumental activities of daily living [IADL]), moderate (need help in activities of daily living [ADL]), or severe (complete dependence for personal care) frailty by CFS [32, 33]. Functional status was measured with ADL (range 0-5; lower scores indicate more dependent) [34] and IADL (range 0-8; lower scores suggest more dependent) [35]. ADL dependence was defined as dependence of having a meal, toileting, bathing, dressing, and transferring from a chair. IADL dependence was defined as dependence of shopping, housework, food preparation, transportation, using telephone, laundry, and handling finances and medication. Fall risk was assessed using the Stop Elderly Accidents, Deaths, and Injuries (scoring as low, moderate, or high risk according to its algorithm) [36]. Cognitive function was evaluated by the Short Portable Mental State Questionnaire (range 0-10; scoring 0-2 imply normal, 3-4 mildly, 5-7 moderately, and 8-10 severely cognitive impairment) [37] and the Confusion Assessment Method (scoring as no confusion or confusion) [38]. Depression status was assessed by the Geriatric Depression Scale-5 Item (range 0-5; scoring ≥ 2 indicate depression) [39]. Nutritional status was assessed using the mini Nutritional Assessment Short Form (range 0-14; scoring 0-7 imply malnutrition, 8-11 at risk, and ≥ 12 normal) [40]. Quality of life was assessed with the EuroQol-5 dimension (scoring as no problem, some problem, or severe problem according to patient’s answer) [41].

 A transdisciplinary meeting was held to plan a personalized care program, focusing on managing geriatric syndromes and improving functional performances.”

“PAC

Patients in both PAC groups received similar services by well-trained geriatric rehabilitation teams. The inclusion criteria for both HPAC and IPAC groups were the same. These teams included geriatric physicians, nurses, physical therapists, occupational therapists, speech therapists, pharmacists, dietitians, social workers, and case managers. Physical therapy emphasized on strengthening exercise, endurance training, balance training, chest care, and ambulation training. Occupational therapy focused on independence in basic and instrumental ADL. Swallowing training, if needed, was provided by speech therapists for those patients with nasal-gastric tube insertion due to an acute swallowing disorder. Nutrition consultation was given by dietitians to patients and their caregivers helping them to prepare daily meals. Social workers provided social resources and welfare information when necessary. In addition, pharmacists played a pivotal role to adjust medication and prevent medication-associated functional decline. The case managers collaborated with team members and patients to fit the program in their daily lives and to enhance their health literacy and ability to self-management. The PAC duration was two to four weeks depending on individual condition.”

“Outcome measurements

The primary outcomes were 90-day ER visits, readmissions, and mortality after PAC. The data were collected from their medical records. Demographics and clinical data including age, gender, medical diagnosis, duration of PAC, rehabilitation sessions, and living environment were also obtained from their medical records. Data were collected by a well-trained research assistant through standard protocols.”

[Comment 2]

Study design – Your inclusion/exclusion criteria should be reported in the body of your paper, not as a supplementary document, where some readers may not have access to them.

[Reply]

Thanks for reviewer’s comments. We have added the inclusion/exclusion criteria in the body of this paper. Please refer to line 116-117; and page 8-9, table 1:

“The inclusion and exclusion criteria of PAC-frailty program were shown in Table 1.”

[Comment 3]

Participants – So some participants remained as inpatients during the intervention? How were they differentiated from “acute” for delivery of “post-acute” care?

[Reply]

We have appreciated the reviewer’s comments. Patients in the inpatient-based PAC group would be discharged from an acute hospital ward and transitioned to a PAC hospital ward. We have revised our statements addressing this concern. Please refer to line 117-120:

“We divided the participants into two groups before being discharged from an acute hospital ward. Patients who participated in the PAC program, considered as intervention group, would be transitioned to their home for home-based (HPAC) or to a PAC hospital ward for inpatient-based (IPAC) model.”

[Comment 4]

Intervention (CGA) – (1) You seem to be describing the PAC instead of the CGA. I suggest describing the assessment separately, and then describing the intervention (PAC) separately. (2) Also, what do you mean by “smooth transition to home” after PAC (line 132)? Isn’t the program delivered after discharge? (3) Please explain where and when the assessment (CGA) and intervention (PAC) were delivered.

[Reply]

Thanks for the reviewer’s comments. 

(1) We have revised the paragraph of CGA and PAC separately according to your suggestions. Please refer to Intervention (CGA) and Intervention (PAC):

“CGA

At the beginning of PAC, CGA was performed at a PAC hospital. The CGA included the following. Participants were identified as mild (need assistances in high older instrumental activities of daily living [IADL]), moderate (need help in activities of daily living [ADL]), or severe (complete dependence for personal care) frailty by CFS [32, 33]. Functional status was measured with ADL (range 0-5; lower scores indicate more dependent) [34] and IADL (range 0-8; lower scores suggest more dependent) [35]. ADL dependence was defined as dependence of having a meal, toileting, bathing, dressing, and transferring from a chair. IADL dependence was defined as dependence of shopping, housework, food preparation, transportation, using telephone, laundry, and handling finances and medication. Fall risk was assessed using the Stop Elderly Accidents, Deaths, and Injuries (scoring as low, moderate, or high risk according to its algorithm) [36]. Cognitive function was evaluated by the Short Portable Mental State Questionnaire (range 0-10; scoring 0-2 imply normal, 3-4 mildly, 5-7 moderately, and 8-10 severely cognitive impairment) [37] and the Confusion Assessment Method (scoring as no confusion or confusion) [38]. Depression status was assessed by the Geriatric Depression Scale-5 Item (range 0-5; scoring ≥ 2 indicate depression) [39]. Nutritional status was assessed using the mini Nutritional Assessment Short Form (range 0-14; scoring 0-7 imply malnutrition, 8-11 at risk, and ≥ 12 normal) [40]. Quality of life was assessed with the EuroQol-5 dimension (scoring as no problem, some problem, or severe problem according to patient’s answer) [41].

 A transdisciplinary meeting was held to plan a personalized care program, focusing on managing geriatric syndromes and improving functional performances.”

“PAC

Patients in both PAC groups received similar services by well-trained geriatric rehabilitation teams. The inclusion criteria for both HPAC and IPAC groups were the same. These teams included geriatric physicians, nurses, physical therapists, occupational therapists, speech therapists, pharmacists, dietitians, social workers, and case managers. Physical therapy emphasized on strengthening exercise, endurance training, balance training, chest care, and ambulation training. Occupational therapy focused on independence in basic and instrumental ADL. Swallowing training, if needed, was provided by speech therapists for those patients with nasal-gastric tube insertion due to an acute swallowing disorder. Nutrition consultation was given by dietitians to patients and their caregivers helping them to prepare daily meals. Social workers provided social resources and welfare information when necessary. In addition, pharmacists played a pivotal role to adjust medication and prevent medication-associated functional decline. The case managers collaborated with team members and patients to fit the program in their daily lives and to enhance their health literacy and ability to self-management. The PAC duration was two to four weeks depending on individual condition.

Patients in the HPAC group received individualized rehabilitation programs at home by a physical therapist, occupational therapist, and/or speech therapist. The program included two to three sessions per week, 50 minutes per session, depending on individual condition. A home-based personalized rehabilitation program, which fitted their daily lives by using their home resources, was provided for patients and their caregivers. The program focused on strength training, ADL and IADL practice, basic mobility training, and reconditioning exercise. Also, individualized meal preparation was taught by a dietitian to the caregivers at home if necessary. The home environment was assessed and modified for those with a high risk of fall. Patients with cognitive impairment, depression, and delirium were referred to neurologists for appropriate treatment. Patients in the HPAC group were followed up at outpatient clinics to manage comorbidities and geriatric syndromes.

A hospital-based geriatric rehabilitation program was provided in the IPAC group at a PAC hospital ward. Rehabilitation was provided by physical therapist, occupational therapist, and/or speech therapist once to twice a day, 50 minutes per session on weekdays. If necessary, physician treated comorbidities and geriatric syndromes. Nursing care was provided for daily vital sign monitor. Other healthcare professionals were consulted during the PAC period as needed. There was no monetary incentive to increase compliance or adherence.”

(2) The “smooth transition to home” meant that patients in the IPAC group would be smoothly transitioned to their home after being discharged from a PAC hospital. They were referred to home healthcare and long-term care service after PAC if needed. We have deleted these sentences to avoid confusion.

(3) The CGA was performed at a PAC hospital at the beginning of the PAC for all participants. The PAC was delivered at either patient’s home or PAC hospital for HPAC and IPAC separately. We have added the statement according to the reviewer’s concern. Please refer to line 131, 166-167, and 177-178:

“At the beginning of PAC, CGA was performed at a PAC hospital.”

“Patients in the HPAC group received individualized rehabilitation programs at home by a physical therapist, occupational therapist, and/or speech therapist.”

“A hospital-based geriatric rehabilitation program was provided in the IPAC group at a PAC hospital ward.”

[Comment 5]

Intervention (PAC) – (1) How long was the program? Your Abstract reported 2-4 weeks, but the average time reported in your Results was only 14 days (2 weeks). (2) Did any participants complete 4 weeks? (3) And if the control was assessed at 2 weeks, would it be a fair comparison to a 4 week intervention? (4) Please describe the intervention in detail. Based on your identification of the therapists, it sounds like a standard rehab program. (5) In particular, what did the speech therapists do? These are frail older adults, not stroke patients. What was the role of speech therapy? (6) Your description of the hospital-based rehabilitation program also sounds like a standard rehab program, but if patients were hospitalized with daily physician rounds, nursing care, etc., how was this “post-acute?”

[Reply]

Thanks for reviewer’s comments. 

(1) The duration of the PAC program is 2 to 4 weeks depending on patient’s rehabilitation goals. Once they achieved their goals, they would be discharged from their PAC program. We have addressed this concern and please refer to line 165:

“The PAC duration was two to four weeks depending on individual condition.”

(2) Only one participant in the PAC group completed the four weeks program.

(3) Although the patients in the control group were assessed at about two weeks, the average observation was 15.3±3.6 which was not significantly different compared to the duration of PAC 14.4±5.4 (p = 0.29). We thought the duration might be fair for both the intervention and control groups.

(4) We have addressed this concern. Please refer to line 154-164:

“These teams included geriatric physicians, nurses, physical therapists, occupational therapists, speech therapists, pharmacists, dietitians, social workers, and case managers. Physical therapy emphasized on strengthening exercise, endurance training, balance training, chest care, and ambulation training. Occupational therapy focused on independence in basic and instrumental ADL. Swallowing training, if needed, was provided by speech therapists for those patients with nasal-gastric tube insertion due to an acute swallowing disorder. Nutrition consultation was given by dietitians to patients and their caregivers helping them to prepare daily meals. Social workers provided social resources and welfare information when necessary. In addition, pharmacists played a pivotal role to adjust medication and prevent medication-associated functional decline. The case managers collaborated with team members and patients to fit the program in their daily lives and to enhance their health literacy and ability to self-management.”

(5) In our program, speech therapists provided swallowing training for those patients with nasal-gastric tube insertion due to an acute swallowing disorder. We have added the statement. Please refer to line 158-159:

“Swallowing training, if needed, was provided by speech therapists for those patients with nasal-gastric tube insertion due to an acute swallowing disorder.”

(6) In this PAC program, physician provided treatments for comorbidities and geriatric syndromes if necessary. Nurse provided daily vital sign monitor. This was different from treatments of the acute medical illness. We have revised the statement according to the reviewer’s concern. Please refer to line 179-181:

“If necessary, physician treated comorbidities and geriatric syndromes. Nursing care was provided for daily vital sign monitor.”

[Comment 6]

Outcome measurements – (1) Please describe the scoring conventions for all of your outcomes. (2) For measures that were categorized for your results (for example frailty was categorized as mild, moderate, severe), please explain how the different categories were determined or calculated. (3) Also, if there are validated Chinese versions of these tools that you used, please cite them.

[Reply]

We have appreciated the reviewer’s comments.

(1) We have addressed the statements according to reviewer’s concern. Please refer to line 132-148:

“Participants were identified as mild (need assistances in high older instrumental activities of daily living [IADL]), moderate (need help in activities of daily living [ADL]), or severe (complete dependence for personal care) frailty by CFS [32, 33]. Functional status was measured with ADL (range 0-5; lower scores indicate more dependent) [34] and IADL (range 0-8; lower scores suggest more dependent) [35]. ADL dependence was defined as dependence of having a meal, toileting, bathing, dressing, and transferring from a chair. IADL dependence was defined as dependence of shopping, housework, food preparation, transportation, using telephone, laundry, and handling finances and medication. Fall risk was assessed using the Stop Elderly Accidents, Deaths, and Injuries (scoring as low, moderate, or high risk according to its algorithm) [36]. Cognitive function was evaluated by the Short Portable Mental State Questionnaire (range 0-10; scoring 0-2 imply normal, 3-4 mildly, 5-7 moderately, and 8-10 severely cognitive impairment) [37] and the Confusion Assessment Method (scoring as no confusion or confusion) [38]. Depression status was assessed by the Geriatric Depression Scale-5 Item (range 0-5; scoring ≥ 2 indicate depression) [39]. Nutritional status was assessed using the mini Nutritional Assessment Short Form (range 0-14; scoring 0-7 imply malnutrition, 8-11 at risk, and ≥ 12 normal) [40]. Quality of life was assessed with the EuroQol-5 dimension (scoring as no problem, some problem, or severe problem according to patient’s answer) [41].”

(2) We have addressed the statements according to reviewer’s concern. Please refer to line 132-134:

“Participants were identified as mild (need assistances in high older instrumental activities of daily living [IADL]), moderate (need help in activities of daily living [ADL]), or severe (complete dependence for personal care) frailty by CFS [32, 33].”

(3) We have cited the validated Chinese version of the tool we used. Please refer to line 132-134:

“Participants were identified as mild (need assistances in high older instrumental activities of daily living [IADL]), moderate (need help in activities of daily living [ADL]), or severe (complete dependence for personal care) frailty by CFS [32, 33].”

Results

[Comment 1]

The participants in your program were quite old, well within the old-old category of aging. I suggest adding their mean age to your Abstract to emphasize the effectiveness of your program.

[Reply]

Thanks for your comments. We have addressed their mean age in the Abstract. Please refer to line 33-34:

“Results: Among 254 participants, 205 (87.6±6.0 years) were in the PAC and 49 (85.2±6.0 years) in the control group.”

[Comment 2]

The data in Table S2 are not effectively reported. What do the variables on the Y-axis represent? For example, how is “frailty” defined? What was improved? Are you saying that 31% of the intervention and 37% of the control improved their frailty in some way? That doesn’t seem to demonstrate that your program was very effective.

[Reply]

We have appreciated the reviewer’s comments and deleted Table S2 to avoid confusion.

[Comment 3]

Table 1 – (1) Please explain either here or in your Methods what the variables along the Y-axis represent. For example, how were mild, moderate, and severe frailty defined? (2) And it appears that on average, most participants continued for the minimum length of the program. If the program was offered for 2-4 weeks, why didn’t they continue for the full 4 weeks? (3) And if the standard deviation for the intervention was 5 days, it appears that some participants didn’t continue for even 2 weeks. Please clarify.

[Reply]

Thanks for your comments.

(1) We have addressed this concern in the Methods. Please refer to line 132-148:

“Participants were identified as mild (need assistances in high older instrumental activities of daily living [IADL]), moderate (need help in activities of daily living [ADL]), or severe (complete dependence for personal care) frailty by CFS [32, 33]. Functional status was measured with ADL (range 0-5; lower scores indicate more dependent) [34] and IADL (range 0-8; lower scores suggest more dependent) [35]. ADL dependence was defined as dependence of having a meal, toileting, bathing, dressing, and transferring from a chair. IADL dependence was defined as dependence of shopping, housework, food preparation, transportation, using telephone, laundry, and handling finances and medication. Fall risk was assessed using the Stop Elderly Accidents, Deaths, and Injuries (scoring as low, moderate, or high risk according to its algorithm) [36]. Cognitive function was evaluated by the Short Portable Mental State Questionnaire (range 0-10; scoring 0-2 imply normal, 3-4 mildly, 5-7 moderately, and 8-10 severely cognitive impairment) [37] and the Confusion Assessment Method (scoring as no confusion or confusion) [38]. Depression status was assessed by the Geriatric Depression Scale-5 Item (range 0-5; scoring ≥ 2 indicate depression) [39]. Nutritional status was assessed using the mini Nutritional Assessment Short Form (range 0-14; scoring 0-7 imply malnutrition, 8-11 at risk, and ≥ 12 normal) [40]. Quality of life was assessed with the EuroQol-5 dimension (scoring as no problem, some problem, or severe problem according to patient’s answer) [41].”

(2) Though the program was offered for 2-4 weeks, the program usually lasted for 2 to 3 weeks. Only if participants needed more time to achieve the goal, the program would be continued for 4 weeks. 

(3) The following were the reasons for those who didn’t continue for even 2 weeks: They achieved the goal earlier than expected; they were discharged from PAC due to personal reasons; their conditions changed and needed further medical intervention.

Discussion

[Comment 1]

It’s difficult to follow your Discussion without a clear and detailed description of the intervention. For example, you say that “transdisciplinary intervention” (line 270) improves health literacy and ability to self-manage, but it’s not clear if your program addressed either of those variables.

[Reply]

We have appreciated the reviewer’s comments and addressed this concern in the Methods. Please refer to line 151-165:

“PAC

Patients in both PAC groups received similar services by well-trained geriatric rehabilitation teams. The inclusion criteria for both HPAC and IPAC groups were the same. These teams included geriatric physicians, nurses, physical therapists, occupational therapists, speech therapists, pharmacists, dietitians, social workers, and case managers. Physical therapy emphasized on strengthening exercise, endurance training, balance training, chest care, and ambulation training. Occupational therapy focused on independence in basic and instrumental ADL. Swallowing training, if needed, was provided by speech therapists for those patients with nasal-gastric tube insertion due to an acute swallowing disorder. Nutrition consultation was given by dietitians to patients and their caregivers helping them to prepare daily meals. Social workers provided social resources and welfare information when necessary. In addition, pharmacists played a pivotal role to adjust medication and prevent medication-associated functional decline. The case managers collaborated with team members and patients to fit the program in their daily lives and to enhance their health literacy and ability to self-management. The PAC duration was two to four weeks depending on individual condition.”

[Comment 2]

Line 271 – What do you mean by “a similar observation?” Similar to what? Did your intervention include nutrition and exercise? You haven’t described your intervention in enough detail to allow readers to compare it to the studies you’re discussing here.

[Reply]

Thanks for the comments. We have addressed the intervention in the Method (PAC), and revised the statements here. Please refer to line 283-286:

“A randomized clinical trial investigated 4 weeks of intervention combining nutrition supplement and exercise rehabilitation for older adults discharged from acute illness reported no significant difference in readmission rates between intervention and control groups [42].”

[Comment 3]

Line 282 – Here you refer back to your exclusion criteria but readers may not have access to your supplemental files. This is an example of why they need to be reported in the body of your paper instead of as supplementary material.

[Reply]

We have reported the inclusion and exclusion criteria in the body of our paper. Please refer to Table 1.

[Comment 4]

Lines 288-290 – Here you make a point about what your program provided but you didn’t describe it as part of your Methods.

[Reply]

Thanks for your reminders. We have addressed our program in the Methods. Please refer to line 150-165:

“PAC

Patients in both PAC groups received similar services by well-trained geriatric rehabilitation teams. The inclusion criteria for both HPAC and IPAC groups were the same. These teams included geriatric physicians, nurses, physical therapists, occupational therapists, speech therapists, pharmacists, dietitians, social workers, and case managers. Physical therapy emphasized on strengthening exercise, endurance training, balance training, chest care, and ambulation training. Occupational therapy focused on independence in basic and instrumental ADL. Swallowing training, if needed, was provided by speech therapists for those patients with nasal-gastric tube insertion due to an acute swallowing disorder. Nutrition consultation was given by dietitians to patients and their caregivers helping them to prepare daily meals. Social workers provided social resources and welfare information when necessary. In addition, pharmacists played a pivotal role to adjust medication and prevent medication-associated functional decline. The case managers collaborated with team members and patients to fit the program in their daily lives and to enhance their health literacy and ability to self-management. The PAC duration was two to four weeks depending on individual condition.”

[Comment 5]

Lines 292-293 – Please define “severe problems” as part of your Methods or your Results.

[Reply]

The severity of self-care or usual activities was according to patient’s answers using the EQ-5D questionnaire (no problem/ some problem/ severe problem). We have addressed this concern in the Methods, and added the statement here. Please refer to line 147-148, and 302-304:

“Quality of life was assessed with the EuroQol-5 dimension (scoring as no problem, some problem, or severe problem according to patient’s answer) [41].”

“In the present study, frail older adults with severe problems in self-care and usual activities according to EQ-5D categories at baseline had a higher risk of adverse outcomes than those without them.”

[Comment 6]

Line 298 – I think you mean “identify” instead of “find out.” You’re making a very important point here, so you want your meaning to be clear. Also, this point may be something to consider as part of your Conclusion.

[Reply]

We have appreciated the reviewer’s comments. We have revised our word according to your suggestion and addressed in the Conclusion. Please refer to line 308-310, and 338-340:

“Therefore, it is important to identify these frail older patients who perceived themselves as having severe problems in self-care and usual activities before discharge from acute hospitalization.”

“Besides, it is important to identify those who reported severe problems in self-care and usual activities to prevent further adverse outcomes.”

[Comment 7]

Line 305 – Your point in this sentence isn’t clear. How can you provide evaluation and discharge planning before acute hospitalization? Do you mean before discharge from acute hospitalization? Please clarify.

[Reply]

Thanks for your reminder. We have revised the statement in line 313-315:

“To provide appropriate transition care, careful evaluation and discharge planning is needed before discharge from acute hospitalization.”

[Comment 8]

Paragraph beginning on line 310 – The study characteristics you identify here should be clearly reported in your Methods. Some of them are either not clear or missing altogether.

[Reply]

Thanks for your comments. We have addressed this concern in our Methods.

[Comment 9]

Line 321 – Excellent point about ethical concerns and randomization.

[Reply]

We have appreciated the reviewer’s comments.

[Comment 10]

Tables – Please go through all of your tables to ensure that each categorical variable along the Y-axis is clearly defined.

[Reply]

Thanks for your reminder. We have addressed each variable in the Methods, Intervention group (CGA). Please refer to line 131-148:

“At the beginning of PAC, CGA was performed at a PAC hospital. The CGA included the following. Participants were identified as mild (need assistances in high older instrumental activities of daily living [IADL]), moderate (need help in activities of daily living [ADL]), or severe (complete dependence for personal care) frailty by CFS [32, 33]. Functional status was measured with ADL (range 0-5; lower scores indicate more dependent) [34] and IADL (range 0-8; lower scores suggest more dependent) [35]. ADL dependence was defined as dependence of having a meal, toileting, bathing, dressing, and transferring from a chair. IADL dependence was defined as dependence of shopping, housework, food preparation, transportation, using telephone, laundry, and handling finances and medication. Fall risk was assessed using the Stop Elderly Accidents, Deaths, and Injuries (scoring as low, moderate, or high risk according to its algorithm) [36]. Cognitive function was evaluated by the Short Portable Mental State Questionnaire (range 0-10; scoring 0-2 imply normal, 3-4 mildly, 5-7 moderately, and 8-10 severely cognitive impairment) [37] and the Confusion Assessment Method (scoring as no confusion or confusion) [38]. Depression status was assessed by the Geriatric Depression Scale-5 Item (range 0-5; scoring ≥ 2 indicate depression) [39]. Nutritional status was assessed using the mini Nutritional Assessment Short Form (range 0-14; scoring 0-7 imply malnutrition, 8-11 at risk, and ≥ 12 normal) [40]. Quality of life was assessed with the EuroQol-5 dimension (scoring as no problem, some problem, or severe problem according to patient’s answer) [41].”

---

## [Decision Letter · Decision Letter 1]

12 Dec 2022

Post-acute care for frail older people decreases 90-day emergency room visits, readmissions and mortality: an interventional study

PONE-D-22-18127R1

Dear Dr. Wu,

We’re pleased to inform you that your manuscript has been judged scientifically suitable for publication and will be formally accepted for publication once it meets all outstanding technical requirements.

Kind regards,

Yoshihiro Fukumoto

Academic Editor

PLOS ONE

Additional Editor Comments (optional):

Reviewers' comments:

Reviewer's Responses to Questions

**Comments to the Author**

1. If the authors have adequately addressed your comments raised in a previous round of review and you feel that this manuscript is now acceptable for publication, you may indicate that here to bypass the “Comments to the Author” section, enter your conflict of interest statement in the “Confidential to Editor” section, and submit your "Accept" recommendation.

Reviewer #1: All comments have been addressed

Reviewer #2: All comments have been addressed

2. Is the manuscript technically sound, and do the data support the conclusions?

Reviewer #1: Yes

Reviewer #2: Yes

3. Has the statistical analysis been performed appropriately and rigorously? 

Reviewer #1: Yes

Reviewer #2: Yes

4. Have the authors made all data underlying the findings in their manuscript fully available?

Reviewer #1: Yes

Reviewer #2: Yes

5. Is the manuscript presented in an intelligible fashion and written in standard English?

Reviewer #1: Yes

Reviewer #2: Yes

6. Review Comments to the Author

Reviewer #1: (No Response)

Reviewer #2: I commend you on the quality and detail of your revisions. Your Methods and Results are now much clearer, and your Discussion makes several quite interesting points.

7. PLOS authors have the option to publish the peer review history of their article (what does this mean?). If published, this will include your full peer review and any attached files.

Reviewer #1: No

Reviewer #2: No

---

## [Editor Report · Acceptance letter]

26 Dec 2022

PONE-D-22-18127R1 

Post-acute care for frail older people decreases 90-day emergency room visits, readmissions and mortality: an interventional study 

Dear Dr. Wu:

I'm pleased to inform you that your manuscript has been deemed suitable for publication in PLOS ONE. Congratulations! Your manuscript is now with our production department. 

Kind regards, 

on behalf of

Dr. Yoshihiro Fukumoto 

Academic Editor

PLOS ONE